



# Technical Note: Flood frequency study using partial duration series coupled with entropy principle

Sonali Swetapadma[1], Chandra Shekhar Prasad Ojha[2]

[1] Research Scholar, Department of Civil Engineering, IIT Roorkee, Roorkee – 247667, Uttarakhand, India
5    [2] Professor, Department of Civil Engineering, IIT Roorkee, Roorkee – 247667, Uttarakhand, India

*Correspondence to*: Sonali Swetapadma (sonaliswetapadma1992@gmail.com)

**Abstract.** Quality discharge measurements and frequency analysis are two major prerequisites for defining a design flood. Flood frequency analysis (FFA) utilizes a comprehensive understanding of the probabilistic behavior of extreme events but has certain limitations regarding the sampling method and choice of distribution models. Entropy as a modern-day tool has 10   found several applications in FFA, mainly in the derivation of probability distributions and their parameter estimation as per the principle of maximum entropy (POME) theory. The present study explores a new dimension to this area of research, where POME theory is applied in the partial duration series (PDS) modeling of FFA to locate the optimum threshold and the respective distribution models. The proposed methodology is applied to the Waimakariri River at the Old Highway Bridge site in New Zealand, as it has one of the best quality discharge data. The catchment also has a history of significant flood events 15   in the last few decades. The degree of fitness of models to the exceedances is compared with the standardized statistical approach followed in literature. Also, the threshold estimated from this study is matched with some previous findings. Various return period quantiles are calculated, and their predictive ability is tested by bootstrap sampling. An overall analysis of results shows that entropy can be also be used as an effective tool for threshold identification in PDS modeling of flood frequency studies.

## 1 Introduction

Frequency analysis of hydrologic events extracts some significant statistical interference from the data that helps in deriving frequency distribution. This distribution becomes a function of the probability of exceedance or return period unique for each gauging site. The at-site flood frequency analysis is suitable for reliably predicting the design discharge of various hydraulic 25   structures to ensure their safety planning and management (Meng et al., 2007; Stedinger et al., 1992). Flood frequency analysis (FFA) comprises two types of sampling approaches: Annual Maximum Series (AMS) and Partial Duration Series (PDS). An annual maximum series includes the largest flow of each year, thereby having one event per year while the partial duration series is derived by extracting all the independent peaks exceeding a particular discharge, called threshold. The average number of events per year ($\lambda$) of a PDS is always greater than the number of years for which data is available (N). So this is beneficial 30   where data are scarce (Lang et al., 1999; Madsen et al., 1997; Önöz & Bayazit, 2001), as it mainly deals with many extreme values comprising primary information about any flood event. A PDS represents the complete flood generating process by





dual modeling of peaks above a threshold, where one is used to model the arrival of peaks and the other for fitting distribution to their magnitude. The application of PDS has some statistical constraints in selecting thresholds and appropriate probability distributions (Guru & Jha, 2016; Adamowski, 2000; Beguería, 2005; Claps & Laio, 2003; Cunnane, 1973; Pham et al., 2014).

Previously, some researchers proposed the identification of thresholds in PDS based on the average number of peaks per year (λ). Langbein (1949) suggested threshold as the lowest annual maximum event of the series, thereby making the value of λ at least one. Similarly, the better performance of PDS with $\lambda$ of 1.65 over the AMS model was observed by Cunnane, 1973; Stedinger et al., 1992; Madsen et al., 1997, etc. Some other studies proposed the choice of threshold depending upon the Poisson arrival of peaks. Cunnane (1979) derived a dispersion index test to check the suitability of the Poisson process in

modeling the arrival rate of peaks. Ashkar and Rousselle (1987) also reported that thresholds should be selected in such a way to make the flood exceedances fit the Poisson process. Following this, Lang et al. (1999) suggested operational guidelines for choosing a threshold where an initial region is identified from the graphical analysis of dispersion index test and the variation of mean exceedances above a threshold and the largest threshold within this region with $\lambda > 2$ or 3 becomes the optimum threshold. Other threshold selection techniques were also proposed; for example, Beguería (2005) applied threshold censoring

with Generalized Pareto and Poisson distribution of PDS modeling. Solari et al. (2017) developed a framework for automatic threshold selection using the Anderson-Darling EDF statistic. Northrop et al. (2017) used the Bayesian cross-validation method to derive inferences from several thresholds instead of finalizing a single threshold value. Some conventional graphical tools also found applications in threshold selection, such as; mean residual life plot (MRLP) and parameter stability plot (Ghosh and Resnick, 2010). Various pieces of literature exist on comparing the performance of PDS with AMS models in flood frequency

analysis. The relatively better performance of PDS compared to AMS even when λ =1 was observed by Bezak et al. (2014). Nagy et al. (2017) also carried out a flood frequency study for the Waimakariri River catchment in New Zealand. Statistical results indicated the better accuracy of PDS over AMS, where the PDS with $\lambda = 3.98$ gave the best results. They suggested the use of PDS is more applicable in those areas where the historical data is unavailable. A detailed review of these threshold estimation techniques and their uncertainty analysis is given by Scarrott and Macdonald (2012). Langousis et al. (2016) also

presented a review of all usual methods available for threshold identification where they classified these approaches into three categories: nonparametric methods, graphical tools, and goodness of fit tests that include statistical metrics and the hill-assumption-based process. They observed that the automation of the mean residual life plot performed better with less sensitivity to the length of the sample and low levels of data quantization.

Besides all these, entropy has emerged as an effective modern-day tool in recent years. It has found a vast application in the

derivation of probability distributions and their parameter estimation based on the principle of maximum entropy theory (POME). For example, Xiong et al. (2018) proposed Halpen distribution with POME for flood frequency study and applied the same to the annual maximum flow series at 12 gauging sites. A Monte Carlo simulation tested the predictive and descriptive ability of the approach. The results suggested that the proposed methodology can be applied as an alternative in FFA. Deng (2019) presented a distribution free method for FFA combining maximum entropy and Akaike's information criterion. Zhang

et al. (2020) applied an entropy based model selection technique in flood frequency analysis with the AMS sampling approach.





Monte Carlo simulation analyzed the performance of the proposed method, which confirmed its better accuracy when the sample size is small with a positive skewness coefficient and bell shaped density function. Even though there exist various threshold identification techniques, there are a very few applications of entropy in PDS sampling of flood frequency analysis. So, the present work augments a new dimension to this field, where an entropy-based approach is proposed to choose the

threshold in PDS as well as the underlying dual models. A new approach based on POME is suggested as a threshold selection criterion. This proposed methodology is applied to the daily discharge data of the Waimakariri River at Old highway bridge site in New Zealand.

## 2 Theoretical background

This section describes the background of the methodology proposed in the present research, which includes (i) probability

distributions for dual modeling of PDS, (ii) the potential of the entropy approach, (iii) entropy functions of probability distributions, (iv) independence criteria and Poisson's hypothesis test, and (v) model selection criteria.

### 2.1 Probability distributions for the dual modeling of PDS

In the present study, four probability distributions are used to model the magnitude of exceedances above a particular threshold: Generalized Pareto distribution (GP), Generalized Extreme Value (GEV) distribution, Pearson type III (P 3), and Log Pearson

3 distribution (LP 3) because of their widespread applications in flood frequency study (Cunnane, 1988; Stedinger et al., 1993, Karim and Chowdhury 1995;  Rao and Hamed 2000; Ghorbani et al. 2010; Chen et al. 2015; Benumar et al. 2017; Drissia et al. 2019; Swetapadma and Ojha, 2020). The shape parameter of such three-parameter distributions considers the effect of skewness present in most hydrologic data series. The Generalized Pareto (GP) distribution is usually known as the 'Peaks Over Thresholds' (POT) model in hydrology as it models the exceedances over the threshold because of its underlying properties

(Davison and Smith 1990; Guru and Jha 2016; Hosking and Wallis 1987; Pham et al. 2014; Smith 1989; and Solari et al. 2017). Generalized extreme value distribution is mainly used to model extreme statistical events. Similarly, various hydrological processes are effectively modeled using the gamma family distributions like P 3 and LP 3 (Bobee and Ashkar, 1991)**.** LP3 distribution is proposed as a standard distribution for design flood estimation in England and Europe (England, 2011; Bezak et al., 2014)**.** The parameters of these distributions are estimated using the L moment method. L-moments are the linear

combination of rank statistics, thereby more robust to outliers in the data than ordinary moments. Also, while estimating quantiles from a small sample, less unbiased inferences can be made using the L-moments (Hosking, 1990; Hosking and Wallis, 1997**;** Sankarasubramanian and Srinivasan, 1999; Bezak et al., 2014). For a sorted sample of length n (such as $x_1 \leq x_2 \leq x_3 \leq x_4 \leq \ldots \ldots \leq x_{n-1} \leq x_n$), the three L moments i.e. $l_1, l_2,$ and $l_3$ can be expressed as,

$l_1 = \beta_0$; $l_2 = 2\beta_1 - \beta_0$ ; and $l_3 = 6\beta_2 - \beta_1 + \beta_0$ ; where $\beta_r = n^{-1} \sum_{i=r+1}^{n} \frac{(i-1)(i-2)\ldots\ldots(i-r)}{(n-1)(n-2)\ldots\ldots(n-r)} x_i$. L skewness ($t_3$) equals to $l_3 / l_2$.

Details of all these distributions, such as their cumulative distribution function (CDF), parameters, and the respective L moment equations, are given in Table 1.





Table 1. Continuous probability distributions used to model the exceedances in PDS (Source: Swetapadma and Ojha, 2020)

| Distribution Models | Cumulative Distribution Function (CDF) | L moment expressions for parameters |
|---|---|---|
| Generalized Extreme Value (GEV) | $F(x) = \exp(-(1-(kz)^{1/k})$ | $C = 2/(3+t_3)$ <br> $k = 7.8590c + 2.9554c^2$ ; $\sigma = \frac{k\lambda_2}{\Gamma(1+k)(1-2^{-k})}$; $\mu = \lambda_1 + \frac{\sigma(\Gamma(1+k)-1)}{k}$ |
| Generalized Pareto (GP) | $F(x) = \dfrac{1}{1-(1+kz)^{-1/k}}$ | $k = (3t_3 - 1)/(1+t_3)$; $\sigma = \lambda_2(1-k)(2-k)$; $\mu = \lambda_1 - \sigma/(1-k)$ |
| Pearson Type III (P 3) | $F(x) = \dfrac{\Gamma_{(x-\gamma)/\beta}(\alpha)}{\Gamma(\alpha)}$ | For $0 < \lvert t_3 \rvert < 1/3$; <br> $z = 3\pi t_3^2$; $\alpha = \frac{1+0.2906z}{z+0.1882z^2+0.0442z^3}$ <br> For $\frac{1}{3} < \lvert t_3 \rvert < 1$; <br> $z = 1 - \lvert t_3 \rvert$; $\alpha = \frac{0.36067z-0.59567z^2+0.25361z^3}{1-2.78861z+2.5609z^2-0.77045z^3}$ <br> For all $t_3$ values; $\beta = \text{sign}(t_3)\, \pi^{1/2}\lambda_2(\Gamma(\alpha)/\Gamma(\alpha+0.5))$, and $\Upsilon = \lambda_1 - (\alpha \times \beta)$ |
| Log Pearson 3 (LP 3) | $F(x) = \dfrac{\Gamma_{(\ln(x)-\gamma)/\beta}(\alpha)}{\Gamma(\alpha)}$ | Same equations as per P 3 distribution |

for GEV and GPA, $z = (x-\mu)/\sigma$; where $k$, $\mu$, and $\sigma$ are the shape, location, and scale parameter respectively. Similarly, $\alpha$, $\Upsilon$ and $\beta$ represent the shape, location, and scale parameters of P 3 and LP 3 distributions.

Based on the dispersion index value, Poisson distribution, Binomial or Negative Binomial distribution is used to represent the arrival of peaks above any threshold (Lang et al., 1999). Table 2 gives details of these distributions, like their probability mass functions and expressions for mean and variance.

Table 2. Discrete distributions used to model the arrival of peaks in the PDS

| Distribution Models | Parameters | Probability Mass Function (PMF) | Mean and Variance |
|---|---|---|---|
| Poisson | $\lambda$ | $P = (\lambda^k e^{-\lambda})/k!$, $k = 0,1,2\ldots$ | $E[X] = Var[X] = \lambda$ |
| Binomial | $n = 0,1,2\ldots$ number of trials <br> $p \in [0,1]$, i.e., success probability of each trial <br> $q = 1-p$ | $\binom{n}{k} p^k q^{n-k}$ | $E[X] = np$ <br> $Var[X] = npq$ |





| Negative Binomial | r > 0; the number of failures until the experiment is stopped $p \in [0,1]$, i.e., the success probability of each trial | $\binom{k+r-1}{k}p^k q^r$ | E[X] = pr/(1-p) Var[X] = pr/(1-p)$^2$ |
|---|---|---|---|

The exceedance probabilities of a PDS and AMS, i.e., $(P(X) = 1 - F(X) = 1 / T)$ are not comparable if $\lambda > 1$. The statistical

relationship proposed by Langbein (1949) based on Poissonian assumption is most commonly used to convert the recurrence intervals from PDS to the annual domain. However, Poisson distribution is not the only choice for modeling the arrival of peaks. So in the present study, the following expression is used (Mohssen, 2009; Nagy et al., 2017).

$$\frac{1}{T_a} = \lambda \left(\frac{1}{T_P}\right)\left(1 - \frac{1}{T_P}\right)^{\lambda - 1} \tag{1}$$

where $T_P$ is the return period in the PDS context and $T_a$ is the annual return period, $1 - F(X) = 1/T_a$.

## 2.2 The potential of the entropy approach

Entropy best describes the unpredictability associated with a system by signifying the amount of disorderness. It is a better measure of information than variance as it relates to higher-order distribution moments (Ebrahimi et al., 1999). C.E. Shannon gave a quantitative measure of entropy for a particular distribution. For 'n' number of discrete random variables such as $Y=\{y_1\ldots \ldots y_n\}$, Shannon's entropy is given by (Shannon, 1948),

$H(y) = E [I(y)] = E [-ln (P(y))]$ (2)

$E$ represents the expected value function, $I(y)$ is a random variable signifying the information contained in the dataset, and $P(y)$ is the probability mass function. The above expression of entropy can be expressed as,

$H(y) = \sum_{i=1}^{n} P(y_i)I(y_i) = -\sum_{i=1}^{n} P(y_i)log_b(P(y_i))$ (3)

Here '$b$' is the base of the logarithm, which defines the units of entropy. In this paper, '$e$' will be used as the logarithm base,

i.e., the unit of $H$ becomes 'Nats'. Similarly, the expression of entropy for a continuous random variable is given below.

$H = -\int_{-\infty}^{\infty} f(y) \ln(f(y)) \, dx$ (4)

$H$ is the amount of uncertainty represented by a probability distribution, and $f(y)$ is the probability density function of the continuous random variable '$Y$'. The above form of entropy is known as 'Continuous entropy' or 'Differential entropy.' Expressions for continuous entropy for various probability distributions can be derived from Eqn. 4.

The principle of maximum entropy given by (Jaynes, 1957) states that while making inferences from limited available data, the probability distribution with the maximum entropy is the best to represent the data. Entropy derives more information from a probability distribution to characterize the input data effectively. So, the minimally biased distribution will have the maximum entropy subject to the available limited data. It will be more probable or less predictable than other distributions with lower entropy values. Therefore, while characterizing unknown events or some limited data with any statistical model,





one should prefer the maximum entropy distribution (Lee et al., 2011). POME has been applied to derive several probability

distributions frequently used in hydrology and their respective parameters (Singh, 1998). Apart from POME, the concept of

entropy has found numerous applications in many areas of research, such as clustering of the homogeneous region ( Basu &

Srinivas, 2013; Yao et al., 2000), thresholding for image edge detection, image grey level thresholding (Chang et al., 1994;

Pal, 1989; Pun, 1981). Some remarkable research in the application of entropy includes (Singh, 1997; Alfonso et al., 2010;

Krstanovic & Singh, 1992; Atieh et al., 2015; Moramarco and Singh, 2010; Hao and Singh, 2011; Rajsekhar et al., 2015;  Li

and Zheng, 2016; Zhang et al., 2020).

**2.3 Entropy functions of probability distributions**

The expression for the entropy of the three-parameter GP distribution is derived here. The probability density function (PDF)

of three-parameter GP distribution is;

$f(x) = \frac{1}{\sigma}(1 + \frac{k(x-\mu)}{\sigma})^{-1-\frac{1}{k}}$ ; for k ≠ 0                    (5)

Entropy for this GP distribution can be derived by putting Eq. (5) in Eq. (4);

$I(f) = ln(\sigma) \int_o^\infty f(x)dx - (-1 - \frac{1}{k}) \int_o^\infty ln(1 + \frac{k(x-\mu)}{\sigma})f(x)dx$                    (6)

Constraints of the equation can be expressed as;

$\int_0^\infty f(x)dx = 1$

$\int_0^\infty ln[1 + k\frac{x-\mu}{\sigma}]f(x)dx = E[ln\left(1 + k\frac{x-\mu}{\sigma}\right)]$                    (7)

So the final interpretation of entropy becomes,

$I_{GP3}(f) = ln(\sigma) - \left(-1 - \frac{1}{k}\right)E\left[ln\left(1 + \frac{k(x-\mu)}{\sigma}\right)\right]$                    (8)

Similarly, the expressions for entropy functions for the other three continuous distributions used in this study can be derived.

For the Generalized extreme value distribution with PDF given as,

$f(x) = \frac{1}{\sigma}(1 - \frac{k(x-\mu)}{\sigma})^{\frac{(1-k)}{k}} exp[-\left(1 - \frac{k(x-\mu)}{\sigma}\right)]^{1/k}$                    (9),

the expression for entropy is derived as,

$I_{GEV}(f) = ln(\sigma) + E\left[\frac{k-1}{k}ln\left(1 - \frac{k(x-\mu)}{\sigma}\right)\right] + E[1 - \frac{k(x-\mu)}{\sigma}]^{1/k}$                    (10)

Similarly, the continuous entropy functions for P 3 and LP 3 distribution are,

$\boldsymbol{I_{P3}(f) = ln(\alpha^\beta \Gamma(\beta)) - \frac{\Upsilon}{\alpha} + \frac{\bar{x}}{\alpha} - (\beta - 1)E[ln(x - \Upsilon)]}$                    (11)

$I_{LP3}(f) = ln(\alpha^\beta \Gamma(\beta)) - \frac{\Upsilon}{\alpha} + (\frac{\alpha+1}{\alpha})\overline{y} - (\beta - 1)E[ln(y) - \Upsilon)],$ y = ln(x)                    (12)

The entropy functions for discrete distributions (Poisson, Binomial, and Negative binomial) can be calculated by simply putting

their probability mass function from Table 2 in Eq. (3).





## 2.4 Independence and Poisson's hypothesis test

One of the basic assumptions of FFA is that the data series to be analyzed is independent or random. If a PDS is not free from
dependent values, it underestimates the variability of the quantiles (Fawcett & Walshaw, 2012). So before conducting any
statistical analysis on a partial duration series, it is essential to justify this independence criterion, which is quite a complex
task. Ashkar and Rousselle (1987) stated that the exceedances above a particular threshold level are independent if the average
return period between successive events is relatively longer. Such a statistical phenomenon cannot merely affect the
independence criteria as the peak discharge values also depend upon various catchment dynamics concerning space and time,
such as catchment area, the frequency of rainfall and their magnitude, etc. (Lang et al., 1999). In the present study, the criteria
given by the United States Water Resources Council (USWRC) are used to select independent peaks above a particular
threshold level. According to which two successive events are independent if they are separated by as many as days as five
plus the natural logarithm of the square miles of drainage area, with the requirement that intermediate flows must drop below
75% of the lower of the two consecutive values (USWRC, 1982). Therefore, two successive flood peaks will be dependent,
which causes rejection of the second peak if they satisfy the following expression.

$$\theta < 5 days + ln\,(A) \text{ OR } q_{min} > (3/4) \, min \, [q_1, q_2] \quad\quad\quad\quad\quad (13)$$

where $\theta$ is the number of days between occurrences of two successive events, $A$ is the catchment area in square miles, $q_{min}$ is
the minimum intermediate discharge between two peaks $q_1$ and $q_2$. The present study applies this independence criterion to
remove all the dependent flood peaks from the PDS derived at each threshold. To justify the independence of these PDS,
modified Kendall's test (Claps and Laio, 2003) is performed at each gauging site. Ferguson et al. (2000) proposed Kendall's
tau test for serial dependence. Visual observation of autocorrelation plots also gives an idea about the independence of peaks.
The Partial Duration Series (PDS) at each threshold is then checked for Poisson's hypothesis by applying the dispersion index
test (Cunnane, 1979), which helps identify the best fit discrete distribution suitable for modeling the arrival rate of peaks per
year. A more detailed description of this test is given by (Lang et al., 1999).

## 2.5 Exceedance model selection criteria

In the present study, different model selection criteria assessed the degree of fitting of continuous distributions to the
exceedances above a threshold. It includes three goodness of fit (GOF) statistics, i.e., Anderson-Darling (AD), modified
Anderson-Darling statistics (ADC), and Kolmogorov-Smirnov test (KS), which measure the fitting of cumulative distribution
functions. However, AD and ADC give more weightage to higher quantiles. Information-based criteria such as modified
Akaike Information Criterion (AICC) and Schwarz Bayesian Criterion (BIC) were also applied as the combination of these
with ADC helps evaluate flood frequency models. Along with this, root mean square error (RMSE), relative root mean square
error (RRMSE), correlation coefficient (CC) were used to measure the error between the observed and predicted quantiles
(Swetapadma and Ojha, 2020). The four candidate distributions were fitted to the magnitude of peaks to measure these





statistical parameters. Based on a statistical ranking method (Olofintoye et al. 2009), these models were ranked between one

to four based on the value of these model selection parameters listed in Table 3. The distribution with the minimum RMSE, RRMSE, AICC, BIC, KS, AD, ADC, or the maximum CC gets the rank one. The ranks assigned from each of these test statistics were added, and the distribution with the lowest total rank became the best fit distribution for the exceedances above a threshold.

Table 3. Model selection criteria for the choice of best fit exceedance distribution (Source: Swetapadma and Ojha, 2020)

| Criteria | Equations | Reference |
|---|---|---|
| Kolmogorov-Smirnov test (KS) | $D = \max_{1 \le i \le n}(F(x_i) - \frac{i-1}{n}, \frac{i}{n} - F(x_i))$ | (Frank and Massey 1951) |
| Anderson-Darling test (AD) | $A^2 = -n - \frac{1}{n}\sum_{i=1}^{n}\{(2i-1)[\ln(F(x_i) + \ln(1 - F(x_{n-i+1}))]\}$ | (Anderson and Darling 1952) |
| Akaike Information Criterion – second-order variant (AICC) | $AIC_c = AIC + \frac{2(m)(m+1)}{n-m-1}$ ; <br> Where $AIC = n \times \ln(RSS/n) + 2k$ | (Burnham and Anderson 2002) |
| Schwarz Bayesian Information Criterion (BIC) | $BIC = (\ln(n) \times k) + (n \times \ln(RSS/n))$ | |
| Root Mean Square Error (RMSE) | $RMSE = [\sum \frac{(O_i - P_i)^2}{n-m}]^{1/2}$ | (Hyndman and Koehler 2006) |
| Relative Root Mean Square Error (RRMSE) | $RRMSE = [\frac{1}{n-m}\sum\{\frac{O_i - P_i}{O_i}\}^2]^{1/2}$ | (Yu et al., 1994) |
| Correlation Coefficient (CC) | $CC = \frac{\sum\{(O_i - \overline{O})(P_i - \overline{P})\}}{\{\sum(O_i - \overline{O})^2 \sum(P_i - \overline{P})^2\}^{1/2}}$ | |
| Modified Anderson-Darling statistics (ADC) | $ADC = \frac{n}{2} - \sum_{i=1}^{n}[(2 - \frac{2i-1}{n})\log(1 - F(x_i) + 2F(x_i)]$ | (Sinclair et al., 1990) |

$f(x_i)$ is the cumulative distribution function; $i$ represents the rank of an observation; $n$ is the length of the sample; m is the number of distribution parameters; RSS stands for the residual sum of squares; $o_i$ and $p_i$ represent the observed and predicted peak discharge values respectively; $\bar{o}$ and $\bar{p}$ are the mean of the observed and predicted series.

**3 Methodology**





There is dual modeling of extreme values in the partial duration series of FFA; one model is used for the arrival of peaks per year ($M_1$), and the other is to fit the magnitude of these peak values ($M_2$). The present study suggests the optimum threshold for PDS analysis is the one where the combined entropy of both these models is the maximum. After removing dependent peaks from the PDS, the variation of the average number of peaks per year and the mean residual life plot are analyzed graphically to identify a suitable range of thresholds. The Dispersion index test gives the appropriate distribution to model the arrival of peaks, and the respective entropy ($HM_1$) is calculated from their probability mass functions. Four candidate distributions are fitted to the magnitude of exceedances to derive the entropy values. The degree of fitting of these continuous distributions to the exceedance series is compared with the conventional statistical approach using eight different model selection criteria described in the previous section. Finally, the total entropy ($H_{total}$) at each threshold is calculated as the sum of these two entropy components. The threshold with the maximum $H_{total}$ is selected as the optimum threshold for PDS modeling of the study area. The optimum threshold derived from the proposed methodology is compared with some existing literature.

The *T* year event is expressed as the *(1-1/λT)* quantile in the PDS perspective. For example, return period estimates ($X_{T,\ predicted}$) are from the GP/PDS model using the following expression (Rosbjerg, 1985)

$$F(x) = 1 - \left[1 + \frac{k(x-\mu)}{\sigma}\right]^{-\frac{1}{k}} = 1 - \frac{1}{\lambda T} \tag{14}$$

where $\lambda$ is the average number of exceedances per year, *T* is the return period (years), and *k*, $\mu$, and $\sigma$ represent the three parameters of the GP distribution obtained from the PDS extracted at a threshold. Similarly, various return period quantiles are computed, and the bootstrap sampling approach helps plot the respective 95% confidence interval.

Figure 1 depicts the detailed methodology followed in this research.





| Preliminary analysis of discharge series |
|---|

⇩

| Extract PDS at those thresholds and apply USWRC independence criteria; check for independence using Modified Mann Kendal's Tau test and auto correlation plots |
|---|

⇩

| Plot t vs $\lambda$ to identify the region where further increase in thresholds cause decrease in $\lambda$ |
|---|

⇩

| Plot mean of exceedance above threshold and identify the region where this mean of exceedances varies linearly with the threshold |
|---|

⇩

| Based on these graphical analysis, identify the range of peak values and apply dispersion index test to find the suitable distribution for modeling the arrival of peaks above that threshold |
|---|

⇩

| Calculate entropy of Model 1 ($HM_1$) (Section 2.3) |
|---|

⇩

| Fit selected continuous probability distributions to the value of exceedances and estimate entropy of all the models ($HM_2$) (Section 2.3) |
|---|

⇩

| Calculate the total entropy at each threshold; identify the optimum threshold as the one with the maximum total entropy ($H_{total}$) |
|---|

⇩

| Comparison of degree of fitness of exceedances with conventional statistical approach using suitable model selection criteria |
|---|

⇩

| Return period flood flow estimation at the optimum threshold considering the underlying distribution models and check for predictive ability through bootstrap sampling |
|---|

**Figure 1: Flowchart showing the detailed methodology followed in this study.**

**4 Study area**

The proposed methodology is applied to the discharge data series obtained for the Waimakariri River at the Old Highway Bridge (OHB) site. The Waimakariri River is one of the largest rivers with 150 km in length and a catchment area of 3654 km$^2$ which flows eastwards from the Southern Alps. It is a large and steep river with a braded gravel-bed river. The upper region





of this catchment is mountainous and glaciated. Flood management is one of the significant issues with the river because of its natural tendency to flow into multiple courses. The major flood in this river is due to heavy rainfall on the Main Divide (Nagy et al., 2017). 30% of the flow is because of snow melting in the spring season (Gray et al., 2006). The Canterbury Regional's Council (Environment Canterbury [ECan] has placed a gauging station on the river at the Old Highway Bridge (OHB). This gauging site has one of the country's excellence and oldest discharge data set. So the quality of data sets available for this site,

along with various studies on the river's flood problem, motivated the authors to apply the proposed methodology here. Hourly data from 1 January 1967 to 31 December 2015 were obtained from Environment Canterbury Regional's Council, and the maximum daily discharge series was extracted to carry out the frequency analysis.

Table 4. Some major flow properties of the data series

| Series | Years available | Mean flow [m³/s] | Standard deviation. [m³/s] | Skewness | Largest flow on the record [m³/s] | Mean annual flood [m³/s] |
|---|---|---|---|---|---|---|
| Waimakariri daily maximum flow | 1967-2015 | 119.064 | 119.379 | 5.576 | 2835.579 in 1979 | 1450.868 |

## 5 Results and discussion

The proposed methodology for the choice of threshold in partial duration series was applied to the daily maximum discharge data for the Waimakariri River at the Old Highway bridge site. The annual maximum series having 49 events was extracted, and initially, some thresholds were applied to derive the respective PDS. Satisfying the independent criteria of the peaks is a prerequisite in any statistical frequency analysis. So, the dependent peaks from those extracted PDS were dropped by following USWRC independence criteria as described in Sect. 2.4 Visual observation of autocorrelation plots confirmed the absence of

serial dependence in the PDS samples. Also, Kendall's Tau test verified the independence of these series, and the PDS at some thresholds were omitted from frequency analysis due to the presence of a significant positive trend at a 95% confidence level.

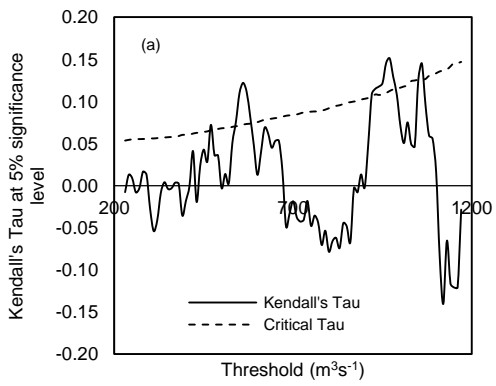

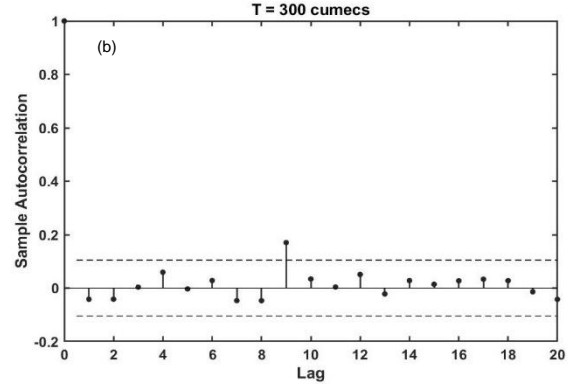

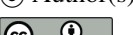



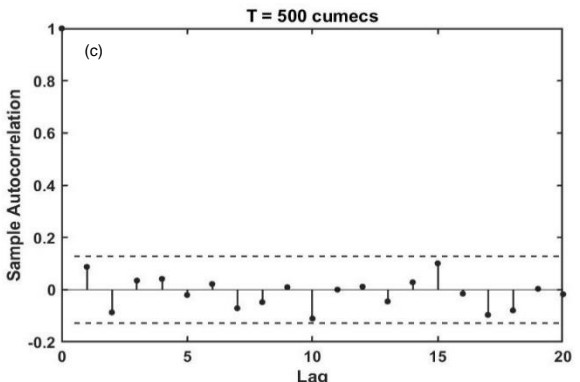
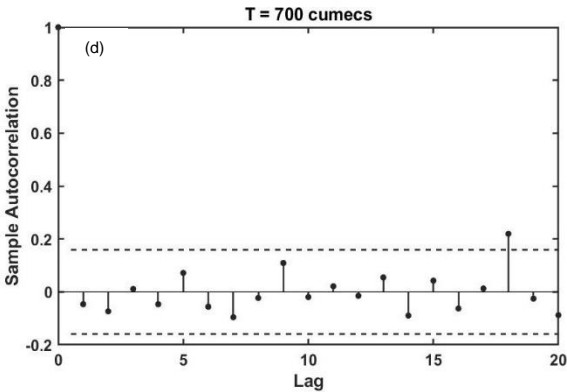

**Figure 2: Test for the independence of flood peaks above the thresholds, (a) Modified Mann-Kendall's Tau test, (b) – (d) autocorrelation plot at a threshold of 300 m³s⁻¹, 500 m³s⁻¹, and 700 m³s⁻¹, respectively.**

For a PDS, an extremely low threshold makes the whole series lie above it, and then with an increase in threshold, more peaks are retained, and the value of $\lambda$ rises. After reaching a peak value, $\lambda$ gradually decreases until no peaks are included when the threshold is greater than the largest discharge in the record. So this gradual variation of the average number of peaks per year divides the entire range of thresholds into four domains, as described by Lang et al. (1999). Figure 3 depicts the variation of the average number of peaks per year ($\lambda$) with threshold level for the study area, and domain 3 was identified.

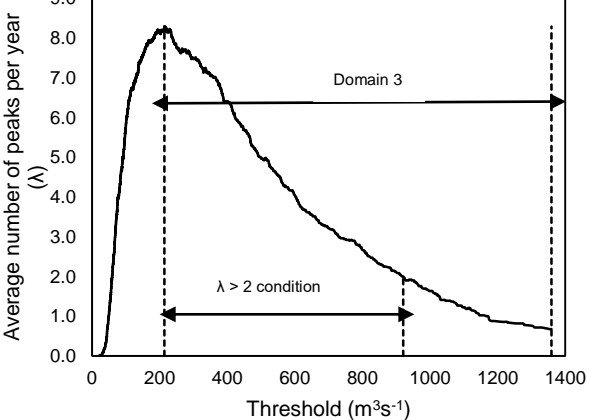

**Figure 3: Variation of the average number of peaks per year with the threshold.**

Lang et al. (1999) also suggested respecting the condition of $\lambda > 2$ while analyzing the variation of $\lambda$. The thresholds with $\lambda > 2$ are marked in Figure 3. Davison and Smith (1990) recommended selecting the threshold within a region where the mean of exceedances above a threshold is a linear function for the stability of the distribution parameters. This plot is known as Mean





Residual Life Plot (MRLP). Figure 4 demonstrates the variation of the mean of exceedances above threshold ($\overline{X_t}$ – t) with the threshold (t) along with the 95% confidence interval of MRLP.

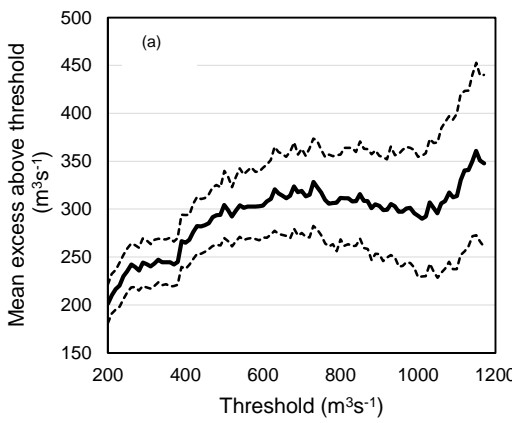 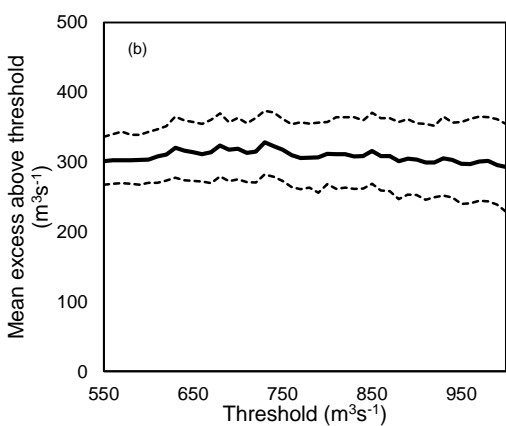


**Figure 4: (a) Variation of the mean of exceedances above the threshold, i.e., MRLP with 95% confidence interval, and (b) A zoomed-in figure of a selected range of thresholds.**

As per MRLP, a threshold should be selected from a region where it shows linear behavior. For the present study area, the PDS extracted between 550 m³/s to 1000 m³/s threshold had a slightly linear pattern, and beyond this, the plot starts to shift.

This change in the graph's linearity with an increase in threshold occurs as the variance of a few extreme values might cause a sudden jump in the plot. So setting an optimal threshold merely based on such graphical observation becomes subjective, but it gives an idea about a range of thresholds where the optimum one may lie. Based on this, thresholds within the range 550 to 1000 m³/s at which PDS satisfied with independence criteria are selected for further analysis.

For the choice of distribution to model the arrival of peaks above any threshold, the dispersion index test proposed by Cunnane

(1979) was applied. Figure 5 displays the value of the dispersion index at a 5% significance level for the study area. The PDS at most of the thresholds follows Poisson's process, with DI lying between the upper and lower limit. The binomial distribution also showed a better fit at some thresholds. Based on this, the entropy of model 1 ($HM_1$) was calculated at each truncation level as described in Sect. 2.3





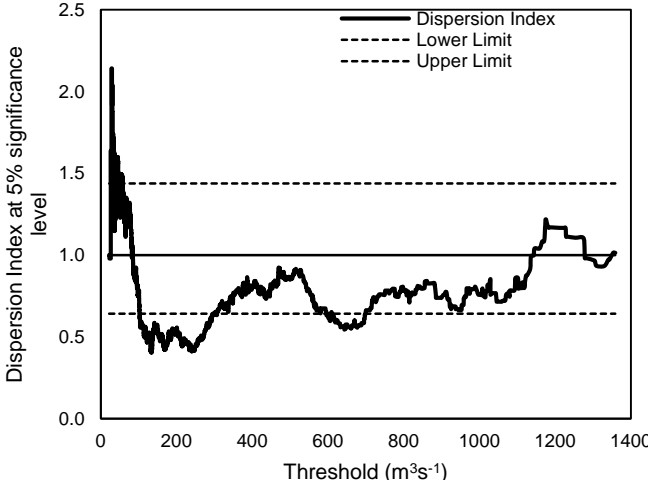

**Figure 5: Dispersion Index test at 5% significance level.**

The four candidate distributions (GEV, GP, P 3, and LP 3) were then fitted to the magnitude of exceedances to compute their entropy function ($HM_2$) as described in Sect. 2.3 The combined entropy of both the models ($H_{total} = HM_1 + HM_2$) was calculated at the chosen thresholds. The threshold with the maximum total entropy was selected as the optimum one for each distribution. Figure 6 demonstrates the variation of entropy functions with the threshold. Figure 6(a-d) compares the total entropy function of individual distributions with the entropy of model 2, i.e. when the distributions were fitted to the magnitude of exceedances.

It's observed that for a particular distribution model, the threshold at which $HM_2$ becomes maximum is different than the threshold at which $H_{total}$ is maximum. For example, GEV has the highest $HM_2$ at 1100 $m^3/s$, while its total entropy reaches the highest at a threshold of 700 $m^3/s$. So consideration of the entropy of model 1 changes the choice of optimum threshold for each distribution.

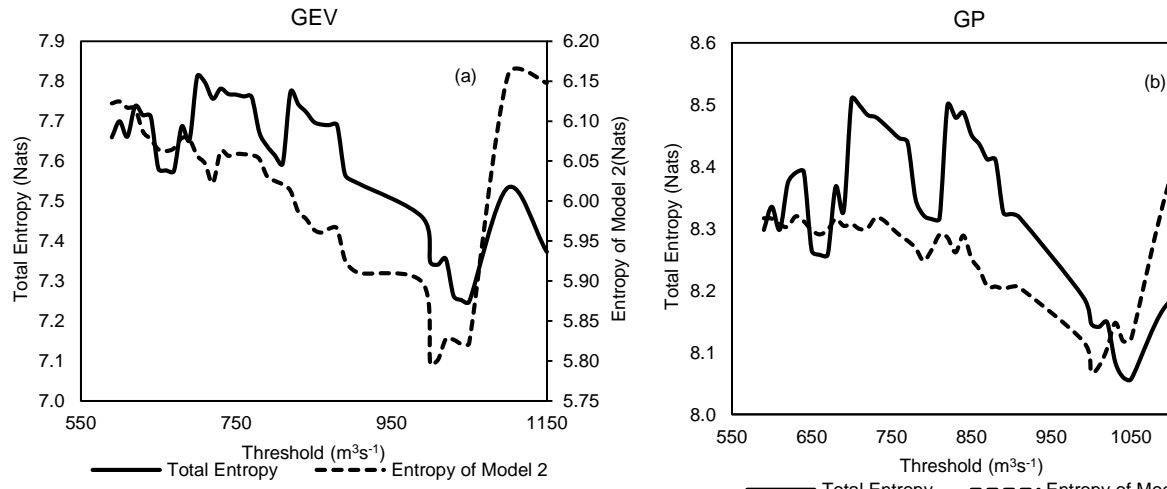






**Figure 6: Variation of entropy with the threshold.**

Figure 6(f) shows the variation of total entropy with thresholds for all four distributions. LP 3 has the maximum entropy at most thresholds, where P 3, GP, and GEV had second, third, and fourth, respectively. LP 3 is recommended as the standard distribution for FFA in the United States by federal agencies (England, 2011). However, the logarithmic conversion of small events in the series may affect the results while using LP 3 distribution. LP 3/PD at a threshold of 710 $m^3$/s was the most suitable choice for PDS modeling of the study area. GP and GEV performed better at 700 $m^3$/s, whereas the PDS at 830 $m^3$/s had the highest total entropy for the P3 distribution. Table 5 summarizes the final results. Poisson's distribution was found to be suitable for the arrival of peaks at these thresholds. The average number of peaks per year varied between 2.5 to 3.2.





Table 5 Summary of optimum threshold and the underlying models

| Distribution models | $T_{opt}$ (m$^3$/s) | $\lambda$ | $H_{total}$ (Nats) |
|---|---|---|---|
| GEV/PD | 700 | 3.22 | 7.812 |
| GP/PD | 700 | 3.22 | 8.510 |
| P 3/PD | 830 | 2.47 | 8.523 |
| LP 3/PD | 710 | 3.18 | 8.756 |

Various test statistics were calculated to check the degree of fitting of these continuous probability distributions at different thresholds for comparing the results of this proposed method to the conventional goodness of fit approaches. So a numerical assessment based on a statistical ranking method was applied using eight model selection criteria described in Sect. 2.5 The distributions were ranked according to these test statistics, and the final rank was computed. Figure 7 represents the rank of these models and their total rank at some thresholds, as an example. A similar analysis was performed at other thresholds also.

GP and LP 3 distribution had better GOF statistics values, i.e., KS and AD at a maximum number of thresholds, implying a better fit of empirical and predicted cumulative distribution function of exceedances. LP 3 distribution had a better combination of modified AD statistics with the information criteria at majority thresholds, which is helpful in flood frequency analysis. Also, the squared error metrics and the correlation coefficient of the exceedance series were better while modeled with LP 3 distribution for most thresholds. Overall, LP 3 distribution performed better for the thresholds lying within 600 m$^3$/s to 1050

m$^3$/s. LP 3 best described the exceedances extracted at 700 and 710 m$^3$/s as per all the test statistics. GP was the second-best model for the exceedances at the majority of thresholds. The results thus obtained agreed with the ones obtained by applying the modified principle of maximum entropy in this research.

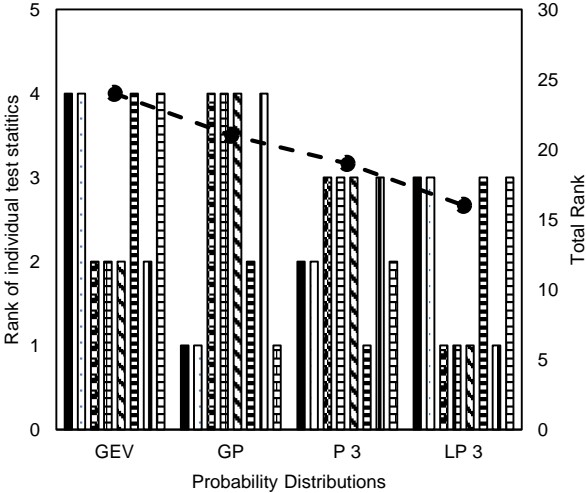

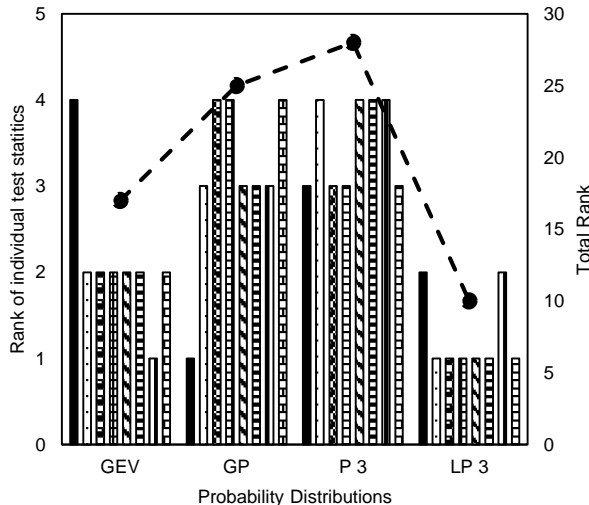





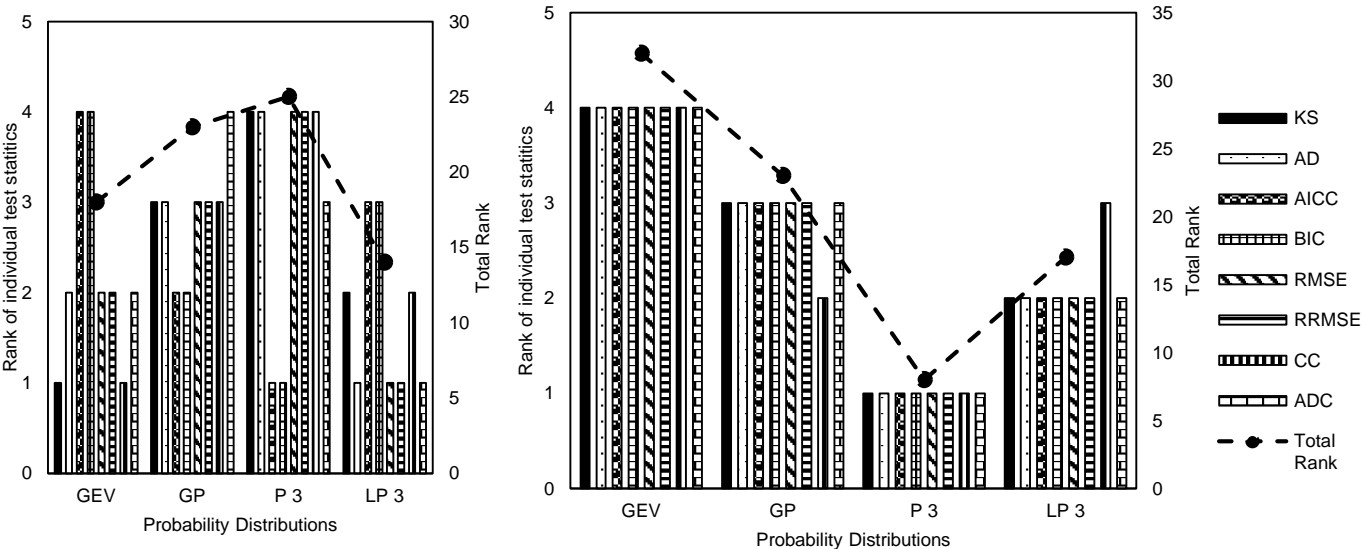

**Figure 7: Ranking of distributions based on eight model selection criteria at a threshold of, (a) 600 m³/s, (b) 700 m³/s, (c) 850 m³/s, and (d) 1100 m³/s.**

10, 50, 100, and 500 year return period estimates were calculated and plotted in Figure 8. GEV and LP 3 distribution models gave higher design flood discharge for $T \geq 50$ years. However, for a lower return period of 10 years, all four distribution models predicted similar design flow values. So the choice of threshold and the respective distribution models don't significantly influence the lower return period estimates. However, for larger quantiles, it plays a vital role. Nagy et al. (2017) also arrived at similar conclusions.

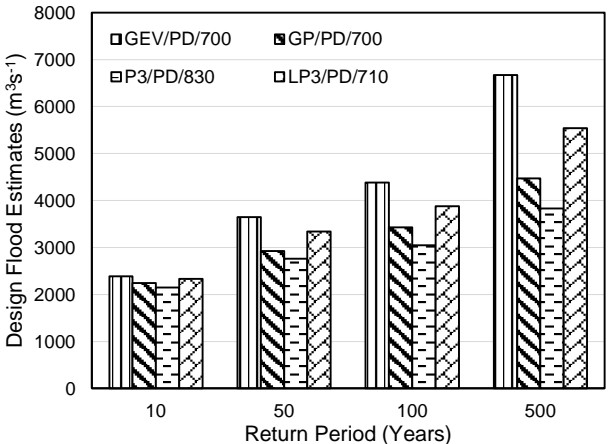

**Figure 8: Quantile estimates of PDS at the optimum threshold.**

A bootstrap sampling was performed with 1000 samples and data length the same as the main PDS to check the predictive ability of these distribution models. The 95% confidence interval (CI) of quantile estimates were plotted and analyzed for



uncertainty. Figure 9 illustrates 95% CI for GP distribution where the estimated flood quantile values lie within the upper and lower limits, thereby justifying the predictive ability of the models.

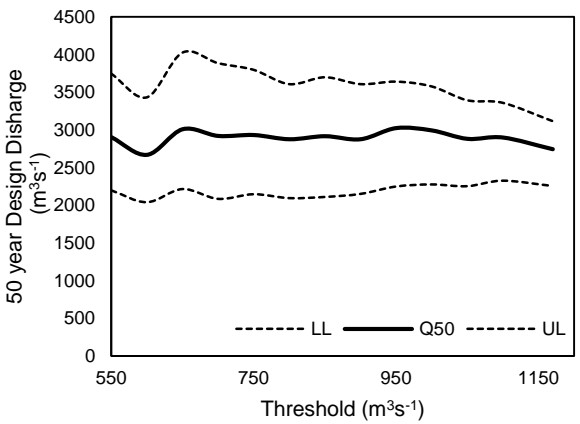 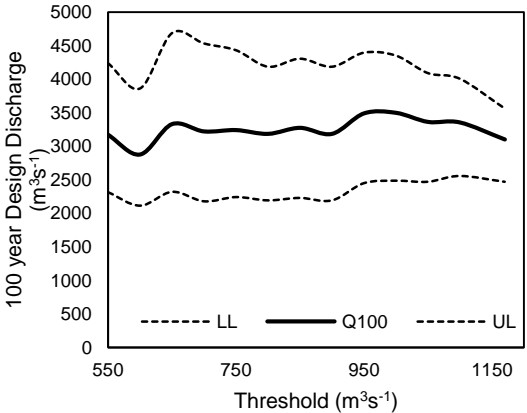

**Figure 9: 95% CI for 50 and 100 year return period quantiles from GP distribution.**

According to the operational guidelines proposed by Lang et al. (1999), the optimum threshold for this study area was identified as 730 m³/s. As per Rosbjerg and Madsen (1992), the threshold from a daily discharge series should be $T_{opt} = E(Q) +$
$3(Var(Q))^{0.5}$, following this, a threshold of 666 m³/s was obtained for the study area. Langbein (1949) stated the threshold as the lowest annual maximum discharge leading to a value of 716 m³/s for the Waimakariri record at OHB. Nagy et al. (2017) calculated a threshold of 700 m³/s for LP 3 and 750 m³/s for GP distribution. It is observed that the optimum threshold value obtained from this present study was close to the findings from some existing threshold selection techniques. Considering the entropy of model 1, i.e., the arrival of peaks instead of taking only the entropy of distributions used for modeling exceedances,
gives more accurate optimum threshold values. The conventional statistical approach ensures only the fitness of models to exceedances; however, the modified POME method helps identify the optimum threshold along with both the models required for describing the PDS. So this new approach of calculating total entropy of dual models of PDS can be used as an alternative to locating the optimum threshold and the respective distribution models.

### 6 Conclusions

Several schools of thought exist regarding the choice of threshold in partial duration series of flood frequency analysis. The present study adds another new domain where the principle of maximum entropy theory is applied to locate the optimum threshold and the underlying distribution models of the PDS. The methodology was applied to the Waimakariri River at OH bridge, New Zealand. After extracting dependent peaks from the PDS, a region of threshold was identified based on the operational guideline proposed by Lang et al. (1999). The dispersion index gave the distribution model for the arrival of peaks
above a threshold, and the corresponding entropy was estimated. All the four candidate distributions were fitted to the



magnitude of peaks to calculate the respective entropy function. The threshold with the maximum total entropy of both these models became the optimum threshold. The fitness of candidate distributions to the exceedances was also compared with the conventional statistical approach, where eight suitable model selection criteria were applied. The results obtained by using POME were similar to the standardized procedure. For all the candidate distributions, the optimum threshold lay between 2.47

to 3.22. The PDS sample with the average number of peaks per year of 3.2 with Log Pearson type 3 and Poisson model performed better. The formula used for converting return periods into annual domain also helped in simplifying the use of PDS by eliminating the compulsory consideration of Poisson's distribution for the occurrence of peaks. Various return period quantiles were estimated, and a bootstrap sampling with 1000 samples resulted in the 95% confidence interval. The results justified the predictive ability of these models derived by applying POME in the PDS context. The threshold obtained in the

present research was close with some previous research. It has an advantage over other existing methods considering both the models while identifying the optimum threshold, i.e., considering the entropy of model 1, i.e., the arrival of peaks instead of taking only the entropy of distributions used for modeling exceedances, gives more accurate optimum threshold values. Overall, the current research suggests this method based on POME in the PDS context as an alternative to the existing conventional approach of threshold selection.

**Data availability:** The hourly discharge data for the Waimakariri River at the Old Highway Bridge gauging site, New Zealand, is available at https://www.ecan.govt.nz/. This work uses material sourced from "the Environment Canterbury Surface Water Archive", which is licensed under a Creative Commons Attributions 4.0 International license by Environment Canterbury.

**Author Contribution:** SS performed data collection, analysis, and manuscript preparation under the proper guidance of CSPO. His intellectual suggestions and review helped in refining the article. Both the authors read and approved the final

manuscript.

**Competing interests:** The authors declare that they have no conflict of interest.

**Acknowledgments:** The authors acknowledge Environment Canterbury Regional Council for providing discharge data for the Waimakariri River at the Old Highway Bridge gauging site, New Zealand.

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
