# Peer review of "Technical Note: Flood frequency study using partial duration series coupled with entropy principle"

_Hydrology and Earth System Sciences, 2021_

## Author Comment (AC1)

The Technical Note "Flood frequency study using partial duration series coupled with entropy principle" by Swetapadma and Ojha discusses methods to use partial duration series type of data to carry out flood frequency estimation. The topic is interesting and appropriate for the journal, but I somehow fail to see what the main contributions of the note, which I think does not provide a clear overview of the new developments, significant advances, and novel aspects of experimental and theoretical methods and techniques which are relevant for scientific investigations within the journal scope (this is a quote from the description of HESS technical notes).

The paper is fairly well organized and the references mostly suitable, giving an overview of what is the current understanding of the question. It presents the modelling framework using a case study in New Zealand.

My understanding is that the novel contribution proposed by the authors is to use entropy as a way to choice the PDS threshold, but I am not entirely sure this innovation is presented in a clear and convincing way. In particular, there are a few points that I find quite unclear or that I believe undermine the strength of the authors' argument: I'll try to outline them below.

1. I feel their note is somewhat lacking a discussion of the consequences connected to the many choices which are done in the modelling pipe-line: the more obvious one to me is the choice of estimating the distribution parameters with L-moments rather than with other methods. Would the threshold/distribution choice be different if we used standard moments or maximum likelihood to estimate the parameters?

In the present study, parameters of the distributions are estimated using L moments, and the reason for the same is described in the manuscript. However, other estimation methods, such as maximum likehood, probability-weighted moments, or method of moments, may be used. Different parameter estimation methods might lead to the selection of different thresholds and distribution choices as the value of entropy may vary accordingly. A similar analysis was carried out by Bezak et al. (2014) where they observed the better performance for the method of L-moments (ML) when compared with the conventional moments and maximum likelihood estimation.

But, all these choices available in the modelling pipe-line will not affect the core of the methodology described in section 3, i.e., the optimum threshold is selected as the one where the total entropy of a PDS model is the maximum. So for any particular parameter estimation technique, the entropy-based methodology proposed in the study will work in the same way, leading to selecting an optimum threshold and the respective distribution models.

2. The choice of distributions used to model the number and magnitude of exceedances could be better motivated. The "traditional" framework uses the Poisson and the Generalized Pareto distribution respectively: these are motivated by some well-known theoretical results. The Negative binomial extends the Poisson distribution, allowing for over dispersion. I do not quite understand how the Binomial distribution is instead fitted here, as we would need to have a k value of exceedances over N "trials" but

the N value should be different from year to year since we only focus on independent peaks. Is this what the authors do? Further the use of the GEV, P3 and LP3 surprised me here as these are typically employed to describe annual maxima and have little theoretical or practical justification in the context of threshold exceedances: they can of course be used, but I'd mention the fact that the GP has a somewhat stronger theoretical grounding.

The Poisson assumption for modeling the number of exceedances above a threshold is the traditional one; however other studies have proposed suitability of Binomial and Negative binomial distributions for the same (Lang et al., 1999; Önöz and Bayazit, 2001; Nagy et al., 2017). As per the Dispersion index (DI) test proposed by Cunnane( 1979) if the value of dispersion index (i) falls within the lower and upper critical DI value ($I_{\alpha/2}$, $I_{1-\alpha/2}$), at a particular significance level α, Poisson process is accepted. If $i < I_{\alpha/2}$, Binomial could be an alternative, and for $i > I_{1-\alpha/2}$, Negative binomial could be used. In the present study, DI at a 5% significance level was calculated at different thresholds and plotted in Figure 5 of the main manuscript. From the plot, it is clear that different distribution models are suitable for different thresholds based on DI value. For the range of thresholds selected from test 1 and test 2 proposed by Lang et al. (1999) (as shown in Figure 3 and Figure 4 of the main manuscript), suitable distribution models such as Poisson and Binomial are chosen from the DI plot. The authors agree with the reviewer that the negative binomial (NB) extends the Poisson distribution, allowing for over-dispersion, and for the study area at some lower value of thresholds, NB was the suitable candidate. However, this range of thresholds was dropped in the further analysis based on Figure 3.

GEV, LP3, P3, and GP distributions are applied to model the magnitude of exceedances in the present study, whereas GP has a more theoretical background. Nagy et al. (2017) carried out PDS modeling of the same study area using similar distributions. So for a better comparison of threshold and distribution models obtained from the present study with their findings, probability distributions from extreme value and Pearson family are also considered in the study along with traditional GP distribution.

3. The definition of AIC and BIC is not correct in Table 3: the definition is, for AIC, n*log-lik(model) + 2k. For the Gaussian case it can be shown that the log-lik of the model reduces to the RSS, but that is a special case of a more general definition. In the caption of the table $o_i$ and $p_i$ should be written using capital letters for consistency with the table content.

For a statistical model with 'k' parameters, AIC developed by (Akaike, 1974) is applied to find suitable probability distribution. It represents the model's lack of fit and unreliability due to the number of parameters. AIC is expressed as, AIC = -2(log maximum likelihood for the model) + 2(number of fitted parameters) = $-2\ln L + 2k$

For 'n' data points assuming the error to be independent identical normally distributed, AIC = $2k + n \ln(RSS/n)$; where RSS is the residual sum of squares. BIC can be expressed as BIC = $n \ln(RSS/n) + k \ln(n)$ with the same assumption of errors obeying Gaussian distribution.

These expressions of metrics have been applied in some previous studies, such as (Karmakar and Simonovic, 2008; Karmakar and Simonovic, 2009; Zhang and Singh, 2007). Based on this literature survey, these definitions of AIC and BIC (Table 3) of the manuscript are applied in the present study.

The necessary corrections are made in the caption of the table.

4. Although the case study is quite interesting I find it is fairly hard to generalize anything from this. How do we know that this approach to PDS modelling is any more suitable than the other currently employed approaches? How could we evaluate that? How does this work in other places? How does this perform under different scenarios of true underlying processes? Overall I think the study does not give enough details about how generalizable the findings are (and actually it is not very clear what the main findings are). The note presents a modelling framework and applies it, but I feel it fails to convince the reader that this modelling approach is somehow better or worth adding to the currently available modelling tools. In particular, I feel the modelling approach as presented is still very much needing the analyst to make some a-priori choices: something that is one of the main issues which make the widespread use of PDS harder to implement.

The present research attempts to propose an alternate method for selecting the optimum threshold in PDS modeling of flood frequency analysis. This method is applied to the daily discharge for the Waimakariri River at the Old Highway Bridge site. The reason being, (i) this is one of the frequently flooded watersheds, and (ii) a recent FFA on this area using the same data series was conducted by Nagy et al. (2017) which provides a base for comparing the results. However, the proposed methodology can also be explored at other sites, which is out of the scope of the recent work. From the literature survey, it's observed that the significant uncertainty in the application of the PDS model in FFA lies in the fact that there is no single method that performs the best for threshold detection in PDS. So the applicability of this proposed entropy-based approach is analyzed only by comparing the value of optimum threshold obtained from some existing guidelines and previous studies.

The authors represent entropy as an alternate tool for threshold selection in the PDS model and found that the threshold obtained from this is close to the other techniques. This is the first study of its kind, where the concept of entropy is applied in PDS modeling of FFA, and also it helps to locate the optimum threshold and the dual models appropriate to model the respective PDS.

**Some other small minor points in the presentation:**

Line 22: interference -> inference

The correction is applied in the revised manuscript.

Line 29: the average number of events can be hardly be larger than the total number of annual maxima. It is often the case that the total number of PDS observations are more than the AMS observations, but this depends on the threshold: a very high threshold might result in PDS which have less observations than AMS.

The required changes are made in line 29.

Line 52: "gave the best results": in what sense? using what metrics? (This is a fundamental question

which might also be addressed in the note: how do we evaluate what methods work well?)

The authors have discussed in detail in the main manuscript (Page number 2) based on which metrics best results were obtained by (Nagy et al., 2017). They analyzed the degree of fitting of PDS samples to the magnitude of exceedances at various thresholds using three statistics such as Chi-square, Kolmogorov-Smirnov (KS), and Filliben Correlation (FCC). They compared the results to find the value of threshold at which PDS has lower $\chi^2$ and KS with higher FCC value.

In the recent work, total entropy at each threshold is compared to find the maximum value. The various error statistics listed in Table 3 are used to evaluate the degree of fitting of the magnitude of exceedances. However, different methods existing in the literature for threshold identification in PDS are compared based on the value of optimum threshold only.

Line 174: "To justify" sounds like an odd wording, maybe "to verify"? Further I would provide some more description of the test (very briefly) specifying the null and alternative hypothesis being tested and how to interpret the result (since these are not really commented on in the text around Figure 2)

The details of these statistical tests are included in Appendix A of the revised manuscript.

Section 5: I would expect somewhere a plot showing the data series

Figure 2 of the revised manuscript represents the daily and annual maximum data series of the study area.

[Figure]

Line 379: the threshold is much higher than 2.47 or 3.22: the threshold which is exceeded on average between 2.47 and 3.22 times per year.

The corresponding changes are made in the revised manuscript.

**References**

Akaike, H.: A New Look at the Statistical Model Identification, IEEE Trans. Automat. Contr., 19(6), 716–723, doi:10.1109/TAC.1974.1100705, 1974.

Bezak, N., Brilly, M. and Šraj, M.: Comparison between the peaks-over-threshold method and the annual maximum method for

flood frequency analysis, Hydrol. Sci. J., 59(5), 959–977, doi:10.1080/02626667.2013.831174, 2014.

Cunnane, C.: A note on the Poisson assumption in partial duration series models, Water Resour. Res., 15(2), 489–494, doi:10.1029/WR015i002p00489, 1979.

Karmakar, S. and Simonovic, S. P.: Bivariate flood frequency analysis: Part 1. Determination of marginals by parametric and nonparametric techniques, J. Flood Risk Manag., 1(4), 190–200, doi:10.1111/j.1753-318x.2008.00022.x, 2008.

Karmakar, S. and Simonovic, S. P.: Bivariate flood frequency analysis. Part 2: A copula-based approach with mixed marginal distributions, J. Flood Risk Manag., 2(1), 32–44, doi:10.1111/j.1753-318X.2009.01020.x, 2009.

Lang, M., Ouarda, T. B. M. J. and Bobe´e, B: Towards operational guidelines for over-threshold modeling, J, Hydrol, 225, 103–117, 1999.

Nagy, B. K., Mohssen, M. and Hughey, K. F. D.: Flood frequency analysis for a braided river catchment in New Zealand: Comparing annual maximum and partial duration series with varying record lengths, J. Hydrol., 547, 365–374, doi:10.1016/j.jhydrol.2017.02.001, 2017.

Önöz, B. and Bayazit, M.: Effect of the occurrence process of the peaks over threshold on the flood estimates, J. Hydrol., 244(1–2), 86–96, doi:10.1016/S0022-1694(01)00330-4, 2001.

Zhang, L. and Singh, V. P.: Trivariate Flood Frequency Analysis Using the Gumbel–Hougaard Copula, J. Hydrol. Eng., 12(4), 431–439, doi:10.1061/(asce)1084-0699(2007)12:4(431), 2007.

---

## Author Comment (AC2)

**General comments:**

The technical note "Flood frequency study using partial duration series coupled with entropy principle" is interesting. The subject is practical in flood frequency analysis. However, I could not determine the tangible advantage of the applied method. From my point of view, this paper is more like a research paper than a technical note. In general, it follows the scopes of the HESS journal. The advantage and novelty of the paper have to be highlighted in the manuscript. Therefore, the manuscript has more room for improvement.

**Specific comments:**

L7 & L14: In the text, you mentioned "quality discharge" several times; what does this term mean?

The authors have mentioned the importance of quality discharge measurement and frequency analysis for effective design flood estimation. The quality discharge signifies the availability of river discharge measurements delivered in real-time. Stream stages and the related discharge determine the hazard level during any flood event. So the availability of the good quality of measured discharge or streamflow data plays a vital role in flood estimation and risk management. The discharge records represent the ground-truth data for developing and continuously improving the hydrologic model's accuracy for forecasting stream flows. So acquiring quality discharge data for streams is critically essential for design flood estimation.

L17-19: In the abstract, more focus on results is needed. Why "POME" is an effective tool?

The authors have modified the abstract by adding the following results from the study.

"For all the four candidate distributions, the average number of peaks per year at the optimum threshold was between 2.47 to 3.22. The PDS sample with λ of 3.2 with Log Pearson type 3 and Poisson model performed better. Also, the results obtained from the proposed method were by the standardized procedure applied using eight model selection criteria."

The principle of maximum entropy given by Jaynes (1957) states that while making inferences from limited available data, the probability distribution with the maximum entropy is the best to represent the data. Such a probability distribution is the "largest one"; it will ignore no possibility, being the most uniform one, subject to the given constraints. This minimally biased distribution will be more probable or less predictable than other distributions with lower entropy values. Therefore, while characterizing unknown events or limited data with any statistical model, one should prefer the maximum entropy distribution. The following lines are added in the abstract to include the effectiveness of the POME tool.

"The POME allows choosing the distribution with the maximum entropy from the set of all probability distributions compatible with one or more mean values of one or more random variables. Initially introduced for solving a problem in statistical mechanics, POME has become a widely applied tool for constructing the probability distribution in statistical inference. Because here the information is generally expressed by mean values of some random variables with a need of a suitable probability distribution which ignores no possibility subject to the relevant constraints."

In the manuscript and especially in the introduction, you used 34 references that are old (before 2000). It is better to employ recent research. However, it is not a critical point.

Some new pieces of literature are added in the modified manuscript, such as Rosbjerg and Madsen (2004), Ben-Zvi (2009), Deidda (2010), Bhunya et al. (2012), Bhunya et al. (2013), Shinyie and Ismail (2012), Caballero-Megido et al. (2018), Pan and Rahman (2021).

L31-33: Could you please elaborate more? How is it possible to thoroughly evaluate the flood generating processes?!

A Partial Duration Series (PDS) compromises traditional time series analysis and Annual Maximum Series (AMS) modeling. It represents more information about any flood event as it involves dual modeling, i.e., the magnitude and time of arrival of peaks above a threshold. Generally, in AMS, we only focus on modeling the maximum value of each year, while in PDS, maximum values higher than a threshold and their rate of occurrence are considered. So it provides a better way to thoroughly evaluate the flood generating process as it incorporates the time of arrival of peaks.

L37: What does "better performance of PDS" mean?

Here the better performance of PDS as compared to AMS signifies less sampling variance of T year return period estimations Q (T) than AMS. For example, Cunnane (1973) observed that the PDS estimate of Q (T) for the same range of return periods has a smaller sampling variance than the AMS estimate only if the PDS has a $\lambda$ of 1.65.

L39: Please explain "Poisson arrival of peaks" before mentioning it in the text.

Poisson arrival of peaks means when the rate of occurrence of peaks above a threshold is suitably modeled using Poisson distribution. The same has been described in the revised manuscript.

L 41: What do you mean by "Poisson process"? Readers demand to have clear fundamental literature in the introduction.

A Poisson process is a model for a series of discrete events where the average time between events is known, but the exact timing of events is random. The arrival of an event is independent of the event before (waiting time between events is memoryless). It is usually used in scenarios where we count the occurrences of certain events that appear to happen at a specific rate but entirely at random. In the case of a Poisson process, events are independent of each other, i.e., the occurrence of one event does not affect the probability another event will occur; the average rate (events per time period) is constant, and two events cannot happen at the same time. The same has been included in the introduction section before presenting the literature on the Poisson model.

Please reflect and indicate your method advantage in the introduction. By having a wide variety of "$\lambda$", what is entropy-based models' preference?

In the traditional statistical approach, the degree of fitness of various probability distributions to model the magnitude of exceedances is compared using some GOF metrics. Based on this, the threshold with better performance of error metrics is selected. The present study is the first of its kind where entropy is applied to locate the optimum threshold in PDS modeling of FFA. Instead of considering only the degree of fitness of magnitude of exceedances, the proposed methodology includes entropy of both the models of PDS, i.e., the arrival rate of peaks and their magnitude, to find the optimum threshold and the respective distributions. The advantage of the proposed method has been included in the introduction section of the revised manuscript. By having a wide variety of '$\lambda$,' the entropy-based model suggests selecting the value of $\lambda$ where the combined entropy of both the models is the

maximum from a region where an increase in threshold causes a decrease in λ value along with linearity in the mean residual life plot.

You mentioned several times "probability dist." And "fitting dist", what are your purposes to point them in the introduction part? I understand what you did, but is it not vivid in your manuscript.

In the introduction, the authors mentioned "fitting distributions", i.e., fitting probability distributions to the magnitude of exceedance in PDS. In PDS, one model is used for the arrival rate of peaks above a threshold and the other for their magnitude.
The authors agree with the reviewer's suggestion, and the same has been taken care of in the revised manuscript.

L66-72: This paragraph must be rewritten to address the purposed method's necessity and novelty. Now, I did not get any points.

Even though several methods exist for threshold identification in PDS modeling of FFA, there is no universal guideline for the same. The present study is the first of its kind, where entropy is applied in PDS modeling. Instead of considering only the degree of fitness of magnitude of exceedances like in the existing standardized statistical approaches, the proposed methodology includes entropy of both the models of PDS, i.e., the arrival rate of peaks and their magnitude to find the optimum threshold and the respective distributions. The proposed methodology is applied to the daily discharge data of the Waimakariri River at the Old highway bridge site in New Zealand. Similar changes are made in the revised manuscript.

L75: In this sentence, what do you mean by "dual"?

Here 'dual' means the two components of a PDS model, i.e., (i) to model the arrival rate of peaks above a threshold and (ii) to model the magnitude of these peaks.

Table 1: What is "Γ"? I did not find its definition in the text.

"Γ" represents the gamma function. It's added in the footnote of Table 1.

L100: What is the benefit of the "negative binomial dist." in your context?

Negative binomial (NB) distribution is an alternate choice of discrete probability distribution to model the arrival rate of peaks in PDS apart from Poisson and Binomial distribution based on the value of dispersion index. From Figure 5, it's clear that NB is suitable for some low range of thresholds. For these values of thresholds, the average number of peaks per year increases with the threshold (Figure 3). So these are excluded for further entropy analysis. However, it might be possible for any other study area to calculate model 1 using NB distribution as it extends the Poisson distribution, allowing for over-dispersion. For such cases, the expression of NB distribution given in Table 1 can be applied in the entropy expression.

L119: Why, in this paper, "e" has to be in a logarithm base?

The logarithm base in the expression entropy defines the unit of entropy. Here, "e" is used as the base in the entire work; however, some other units can also be used as they won't affect the core of the methodology proposed in the study.

L129-130: "therefore, while … (Lee et al., 2011). I do not understand this sentence.

Here the importance of the principle of maximum entropy (POME) theory is described as observed by Lee et al. (2011). For a statistical model with limited available data, the distribution with the maximum entropy should be chosen as it will be more probable or less predictable than other distributions with smaller entropy values.

L131-136: irrelevant to previous sentences; it has to be somewhere else.

This section represents some literature on applying the POME theory, which has now been shifted to the introduction section in the revised manuscript.

L163-165: Rewrite the sentences. It is not understandable.

"The peak discharge in a PDS also depends upon various catchment dynamics with respect to space and time, such as catchment area, the frequency of rainfall and their magnitude, etc. So the independence criteria of peaks can depend not only on statistical phenomenon as proposed in some previous literature."

L169-171: Could you please graphically explain this condition? then "intermediate" discharge can be intelligible in L173.

[Figure]

This graph is added to the revised manuscript.

L171: Do you mean mathematically and logically using OR in Eq. 13? Because it has to be in this form.

$\theta < 5$days $+ \ln (A)$ **or** $q_{min} > (3/4)$ min $[q_1, q_2]$; this is the right expression.

Table 3: ADC has to be mentioned after AD, not in the end.

The changes are made in the revised manuscript.

L200-201: Does not have vivid meaning.

As described previously, the PDS extracted at any threshold comprises two models; a discrete distribution is used to model the arrival of peaks per year (M1). A continuous probability distribution fits the magnitude of these peak

values ($M_2$). The present study suggests selecting a threshold where both models' combined entropy is the maximum as per the principle of maximum entropy theory.

L206: What is conventional statistics in this research? And what do you want to point by comparing these models?

The conventional statistics used in this research are listed in Table 3. The standardized statiscal procedure found in literature evaluates the degree of fitting of various probability distribution models to the magnitude of exceedances applying one or more such statistical measures. So the authors have compared the result obtained from the proposed methodology (overall degree of fitting of candidate distributions at various thresholds) with this conventional standardized procedure to analyze whether this entropy-based analysis can be applied as an alternate for threshold selection.

Page 9: What does this method work if λ=2? Two independent events per year. Is there any way to calculate the threshold by assuming two events per year?

$\lambda = n/N$; where 'n' is the total number of peaks above the threshold and N is the number of years available data. Here, N = 49, so the threshold at which there are 98 peaks (n) above it corresponds to a $\lambda = 2$. So for Figure 4, the independence criteria are applied to extract the PDS at each threshold starting from the minimum daily discharge (22.033$m^3$/s). This analysis shows that at t = 920.368 $m^3$/s, there are 98 peaks above it, referring to $\lambda = 2$.

Figure 1: It is good instruction; however, you need to elaborate more on the second box "Extract PDS at …", explain the third box in the text, before this figure, and answer the question of "what if for non-linear approach" for the fourth box.

The second box: Extract PDS at all the thresholds starting from the minimum daily discharge and apply USWRC independence criteria; check for independence using Modified Mann Kendal's Tau test and autocorrelation plots

An explanation for the third box (before Figure 2): The gradual variation of the average number of peaks per year divides the entire range of thresholds into four domains, as described by Lang et al. (1999). For a PDS, an extremely low threshold makes the whole series lie above it in domain 1. Then with an increase in threshold, more peaks are identified and retained in domain 2. Further rise in threshold makes a decrease in the average number of peaks per year in domain 3, and finally, when the threshold reaches the time series maximum, no peaks are retained in domain 4.

Fourth Box: Linearity of MRLP gives us a rough idea about a range of thresholds where the optimum threshold might lie. It also implies choosing a threshold to maximize the stability of the distribution parameter estimates for the PDS. If for some cases, no region of linear variation is found in the MRLP, the independent thresholds from Domain 3 (Figure 4) can be considered further. However, the MRLP and parameter stability plot should be analyzed from the bootstrap sampling to ensure the same.

L 225-226: remove it. It is not relevant. "Flood management …"

The lines are removed in the revised manuscript.

L228: What does excellence mean in FFA? Does it mean long-term?

Here the excellence of the data set means the long–term available data and its quality (with very few missing values).

Table 4: The mean and std of maximum daily flow are the same. I do not have your data. Do you think, is it correct?!

There was a mistake in the calculation. The values have been modified in the revised manuscript.

L238: What are the applied thresholds? Please write them in this part.

Initially, each data point of the daily data series starting from the minimum daily discharge (22.033m$^3$/s) is considered a threshold. USWRC independence criteria removed the dependent peaks (Figure 3). After identifying a suitable range for the threshold from the graphical tests proposed by (Lang et al., 1999), i.e., in Figure 3, arbitrary thresholds are selected within that range at an interval of 10 m$^3$/s such as 220 m$^3$/s,230 m$^3$/s,240 m$^3$/s …….1150 m$^3$/s. Similar explanations have been included in the revised manuscript.

Figure 2: in a, did you omit the values upper the critical dash line? What is the interpretation of the negative values in Kendal tau (y-axis)?

The thresholds at which tau is greater than tau critical (upper dashed line) are omitted from further analysis as it violates the independence assumption of peaks. The negative Kendal tau value represents a negative correlation among the variables.

I am curious to know the reason for the higher correlation in 9 step time lag in b.

Figure 2b represents the autocorrelation plot for the PDS extracted at a threshold of 300 m$^3$/s. The partial duration series is shown below.

[Figure]

The authors have used MATLAB software to plot the autocorrelation graph using autorcorr() function, and the result is shown below.

[Figure]

The PDS graph shows that except for two higher discharges, the peaks extracted at a threshold (T) of 300 m³/s do not have any particular pattern, i.e., they tend to fluctuate randomly. The presence of these two higher values at 9 step time lag might be the reason for a higher correlation. But since most of the spikes are not statistically significant, this implies that the peaks are mainly independent of each other.

L244: How many peaks did you select in the designated threshold?

Only independent peaks are selected at each threshold, those retained after applying USWRC criteria.

Figure3: is an excellent figure, but a question rase up, why did you consider values below λ=1? What is the benefit of showing, for example, eight peaks per year in FFA? Because they are not "flood" anymore.

Here to analyze the graphical test proposed by Lang et al. (1999), we have considered all the peaks starting from the lowest daily discharge value. The number of independent peaks in the PDS and the respective λ values are calculated at each point. So it also includes the values below λ = 1 to identify all the four domains of the plot. Eight peaks per year are not shown for any particular reason; it's just the part of the entire plot of t vs. λ as λ attains a maximum of 8.22. This represents the lower limit of domain 3.

L278-279: How do you recognize the linear behavior in the figure? This is an entirely ocular and non-mathematical diagnosis. Where does the plot start to shift?! How do you consider linearity if the threshold in Figure 4b? By changing the y-axis, it is not linear anymore!

The authors agree with the reviewer that it's based on visual observation. It's observed from the calculated values that for a threshold varying between 220 m³/s to 1150 m³/s, the mean excess varied between 216 to 360 m³/s. So the authors have identified a region where the value of mean excess doesn't show much variation with change in the threshold, i.e., within a threshold of 550 to 1000 m³/s, mean excess showed a slight variation around 300 m³/s. The Y-axis of Figure 4b is made the same as Figure 4a in the revised manuscript.

L283: explain more.

A similar explanation given in the previous question is added in the revised manuscript.

Page 14: What is the range of dispersion index? What does 1 in your study mean? What is/are the reasons for having high DI at low thresholds? and reflect it in the manuscript.

Dispersion index can take any positive value greater than one or less than or equal to 1. For Poisson's process, mean equals to the variance lead to DI of 1. More the line comes close to the line DI = 1, Poisson's hypothesis becomes more applicable. In the present study, DI values are plotted to identify suitable probability distribution to model the arrival rate of peaks. It's unnecessary to highlight the DI = 1 line in our context. So the required changes are made in the revised manuscript. At low threshold values, more peaks above the thresholds cause over-dispersion, leading to high DI values.

L319: "The average number of peaks per … 2.5 to 3.2", It is a wide range for long-term high-resolution time series. What do you think about that? Could you suggest λ=3 as an average value for your case study area? Or, it is still sensitive to this range.

Here, the authors expressed the average number of peaks per year of the four candidate distributions at their respective optimum thresholds (Table 5, column 3). For P3/PD, the optimum threshold has a λ of 2.47 while GEV/PD and GP/PD λ equals 3.22. "The average number of peaks per year varied between 2.5 to 3.2": this line was not the correct representation. So it's been modified in the revised manuscript.

Table 5: It is not needed to write the λ column here. It is not in the continuation of the entropy section.

The same has been removed from Table 5.

L320-324: Do not need to mention here.

These lines are removed from here as they have already been discussed in the methodology section.

L324: Rewrite the sentence "A similar analysis …"

"The degree of fitting of the four candidate distributions were assessed and ranked according to the test statistics (Table 3). The final rank was computed at the selected thresholds as described in the methodology section."

L339-340: Did you have any other expectation for having a higher design flood for the considerable return period in GEV?!

The authors did not find any apparent reasons for having a higher design flood for a considerable return period in GEV.

Figure 9: What is the reason for the abrupt jump around 950? I know, what is LL & UL but please mention their abbr.

The PDS extracted at $950 m^3/s$ contains some dependent peaks (tau=0.122, tau critical = 0.1082). So the presence of these peaks might have caused an abrupt jump around 950. LL and UL represent the lower and upper limit at 95% confidence level for the bootstrap sampling. Similar changes are made in the revised manuscript.

L359: Why and how do you select 730?

According to the guidelines proposed by (Lang et al., 1999), (1) identify an interval of threshold values which gives good results for tests nos. 2 and 3 (Figure 4 and Figure 5 in the present study); (2) select within this interval the largest threshold with $\lambda > 2$ or 3 (test no. 1) (Figure 3). Based on this, 730 is selected.

Page 17: Still, I do not understand the advantages of your method!. Is it faster? Is it hydrologically more reasonable? Is it prevent to do some additional steps?

Instead of applying several statistical measures to assess the degree of fitness of models to the value of exceedances, only this combined entropy of both the models proposed in the study can lead to similar results of the optimum threshold. It's hydrologically more reasonable as it considers the entropy of both the models of a PDS, i.e., the uncertainty involved in both the models while finalizing the threshold. The authors have explored the application of entropy in PDS modeling and proposed this POME-based approach as an alternate for other threshold identification techniques.

Conclusion question: Is it possible to have no peaks per year? I mean, the average peak per year is 3.2, and theoretically, it is possible to have several independent peaks in a year and no peak in another year. Did you have such a drought year or period?

Yes, it's possible to have a dry year, i.e., no peaks are above the selected threshold in a particular year. Yes, it's possible for any value of $\lambda$. Such drought years were observed at some threshold in our study also.

**Technical corrections:** The authors would like to thank the reviewer for suggesting these technical corrections, and the required changes are done in the revised manuscript.

L36: You already mentioned "$\lambda$" in the text.

L46: Please define EDF abbr.

EDF stands for Empirical Distribution Function.

L93-94: Please write this part in equation format.

L93-94 is written in equation format in the revised manuscript.

Table 1: Please cross-check the L moment expressions. I believe it has a mistake in the typing.

The expression is corrected.
$C = 2/(3+t_3) - (\ln2/\ln3)$

L98: GPA is wrong. It is GP all over the manuscript.

The GPA is changed to GP in L98.

L 109: Can be merged to the above equation.

It's merged with the equation written above.

Page 5: "y" and others are not the same format as other parts of the paper. i.e., "y" à y

L 114: by (Shannon, 1948) is the wrong citation form.

L 141: Eq. (4), while in line 124, it is written Eqn. So it has to be the same in all parts of the text.

L 149: the "Generalized extreme value" should be "Generalized Extreme Value" or "generalized extreme value".

"Generalized Extreme Value"

L154: Why eq. 11 is bold?

L177: The repetition of (PDS) is not needed anymore.

L185: When you mention the "Schwarz bayesian criterion" instead of the "Bayesian information criterion" term, you should write SBC, SIC, or SBIC, not BIC.

The corrections are done in the revised manuscript.

L203: "The Dispersion ind." Should be "The dispersion ind.".

It's been changed to "The dispersion index".

L228: Extra parenthesis

It is removed in the revised manuscript.

L239: in Sect. 2.4 ---à in Sect. 2.4.

Figure 3: Please fix the place of the arrow for the domain3.

The same has been corrected.

Figure 4: It is better to have the same x-axis (200-400). Also, y-ais in b is not appropriate.

The required change in Figure 4 is done.

L284-285: by Cunnane is mentioned several times.

L286: DI, did you mention this abbr in the text before?

Please take care of using abbr in the text. Sometimes, it seems that they are written too much!!

DI stands for Dispersion Index. All such abbreviation-related mistakes are modified in the revised manuscript.

Figure6: Different y-axis makes it difficult to compare total entropies. You can at least use the same minor grid with two decimals.

Using colors may be better to show the result. Sometimes it is not easy to recognize the exact points.

Figure 6 is modified with colors and the same range of the Y-axis.

L325: KS and AD statistics

Figure 7&8: Surely use colors. Legends are not readable for me.

Figure 7 and Figure 8 are modified accordingly.

L373: Different abbr. at OH. Sometimes it is OBH.

The same abbr. OHB (for the Old Highway Bridge site) has been updated in the revised manuscript.

**References**

Ben-Zvi, A.: Rainfall intensity-duration-frequency relationships derived from large partial duration series, J. Hydrol., 367(1–2), 104–114, doi:10.1016/j.jhydrol.2009.01.007, 2009.

Bhunya, P. K., Singh, R. D., Berndtsson, R. and Panda, S. N.: Flood analysis using generalized logistic models in partial duration series, J. Hydrol., 420–421, 59–71, doi:https://doi.org/10.1016/j.jhydrol.2011.11.037, 2012.

Bhunya, P. K., Berndtsson, R., Jain, S. K. and Kumar, R.: Flood analysis using negative binomial and Generalized Pareto models in partial duration series (PDS), J. Hydrol., 497, 121–132, doi:https://doi.org/10.1016/j.jhydrol.2013.05.047, 2013.

Caballero-Megido, C., Hillier, J., Wyncoll, D., Bosher, L. and Gouldby, B.: Technical note: comparison of methods for threshold selection for extreme sea levels, J. Flood Risk Manag., 11, 127–140, doi:https://doi.org/10.1111/jfr3.12296, 2018.

Cunnane, C.: A particular comparison of annual maxima and partial duration series methods of flood frequency prediction, J. Hydrol., 18(3–4), 257–271, doi:10.1016/0022-1694(73)90051-6, 1973.

Deidda, R.: A multiple threshold method for fitting the generalized Pareto distribution to rainfall time series, Hydrol. Earth Syst. Sci., 14(12), 2559–2575, doi:10.5194/hess-14-2559-2010, 2010.

Jaynes, E. T.: Information theory and Statitical mechanics,  Phys. Rev., 106(4), 620-630, 1957.

Lang, M., Ouarda, T. B. M. J. and Bobe´e, B: Towards operational guidelines for over-threshold modeling, J, Hydrol, 225, 103–117, 1999.

Lee, S., Vonta, I. and Karagrigoriou, A.: A maximum entropy type test of fit, Comput. Stat. Data Anal., 55(9), 2635–2643, doi:10.1016/j.csda.2011.03.012, 2011.

Pan, X. and Rahman, A.: Comparison of annual maximum and peaks-over-threshold methods with automated threshold selection in flood frequency analysis: a case study for Australia, Nat. Hazards, doi:https://doi.org/10.1007/s11069-021-05092-y, 2021.

Rosbjerg, D. and Madsen, H.: Advanced approaches in PDS/POT modeling of extreme hydrological events, in

Hydrology: Science and Practice for 21st Century, Proceedings of the British Hydrological Society International Conference, edited by B. Webb, N. Arnell, C. Onof, N. MacIntyre, R. Gurney, and C. Kirby, pp. 217–220, Imperial College, London, U. K., 2004.

Shinyie, W. L. and Ismail, N.: Analysis of T-year return level for partial duration rainfall series, Sains Malaysiana, 41(11), 1389–1401, 2012.

---

## Author Comment (AC3)

The manuscript presents an approach based on entropy for choosing the most suitable statistical models to represent partial duration series of streamflow. In particular, it proposes to evaluate the combined entropy of the statistical models describing the arrival of peaks above a certain threshold and the magnitudes above this threshold.

The idea is interesting, especially because it advocates using additional criteria (i.e., the capability to represent occurrences of events exceeding a threshold) to the goodness of fit of theoretical distribution calibrated to magnitude exceedances above the threshold. However, the study has some issues which prevent reaching substantial conclusions. They are described below.

- Calculating entropy constitutes an additional step to the usually applied procedure in this field. The value of performing this additional step should be made clear. As I stated above, I see value in the evaluation of an additional criterion to the goodness of fit of theoretical distribution calibrated to magnitude exceedances above the threshold. However, what advantage does it actually bring with it? Does this method for choosing the most suitable statistical model improve its predictive power? The authors claim it does, but the support of this claim is not clear to me (see the next comment).

In the present study, the authors have proposed entropy as alternate goodness of fit (GOF) measure for threshold identification in partial duration sampling of flood frequency analysis. The prerequisites for threshold identification in PDS, such as satisfying the graphical tests proposed by Lang et al. (1999) and independence of peaks by several guidelines or other statistical tests, remain the same. The next step of evaluating the degree of fitness of different distributions to model the magnitude of exceedances generally involves the application of several GOF measures such as Chi-squared, Kolmogorov-Smirnov (KS) and Filliben Correlation (FCC), Anderson-Darling (AD), modified AD, root mean square error (RMSE), mean absolute error (MAE), relative mean absolute error (RMAE), different information criterion, etc. The present study recommends maximizing the total entropy of both the models of the PDS instead of calculating different combinations of error criteria as suggested in several pieces of literature. So the calculation of entropy has been proposed as an alternate step in threshold identification of PDS in flood frequency analysis. To this end, the authors have compared results attained from the proposed methodology with those obtained applying combinations of several error statistics (Table 3 and Figure 7 of the manuscript) for the study area considered in this work. LP 3 has the maximum total entropy at most thresholds, where P3, GP, and GEV possess the second, third, and fourth, respectively (Figure 6f). Similar results are obtained from the statistical ranking of distributions also (Figure 7). Comparable results are observed for individual distributions, such as LP 3 has the maximum total entropy at a threshold of 710 m3/s. It best described the exceedances extracted at 700 and 710 $m^3$/s as per other test statistic's rankings.

The study involves the evaluation of an additional criterion to the goodness of fit of theoretical distribution calibrated to magnitude exceedances above the threshold. It's observed that the threshold with the maximum entropy of Model 2 (i.e., used for the magnitude of exceedances) is different from the one at which the total entropy of both the models is the maximum (Figure 6). The latter, selected as the optimum threshold, is close to other values obtained from existing literature. So combining entropy of Model 1 (arrival rate of peaks) improves the accuracy of threshold identification. For example, the entropy of Model 2 is the maximum at a threshold of 900 $m^3$/s for LP 3 distribution, while combining entropy of both the models maximizes entropy at 710$m^3$/s, which is similar to the optimum thresholds found from the existing guidelines and studies carried out in the study area previously. Also, the optimum threshold obtained from the proposed method lies in the stabilized region of Figure 9, which justifies the predictive ability of the model.

- The authors claim to discuss the predictive ability of the statistical model selected by means of the proposed entropy-based approach in Figure 9. The figure shows the flow value associated to 50 and 100 years return period, calculated by means of a generalized Pareto distribution calibrated to exceedances above a set of different thresholds. Confidence intervals of the estimates are also displayed. I do not understand what this figure tells about predictive ability. I would be happy to hear about it; in case I am missing something obvious. First of all, Log Pearson 3 is the most suitable statistical distribution according to the values of entropy, whereas results for generalized Pareto are shown here. Then, where do we see in Figure 9a better predictive performance of the distribution suggested by the entropy metric? Also, its predictive performance is better compared to what? I guess it should be better compared to the performance of the statistical model that would have been chosen based on goodness-of-fit metrics displayed in Figure 7 (see the next comment about the interpretation of those results).

  In bootstrapping, we repeatedly sample from the observed dataset, with replacement, forming a large number (B=1000 in this study) of bootstrap datasets, each of the same sizes as the original data. The idea is that the actual observed data takes the place of the population of interest, and the bootstrap samples represent samples from that population. To use bootstrapping for analyzing the predictive ability of an estimate, we fit our model to the original data and fit the model to each of the B bootstrap sample datasets. We calculate values for 95% confidence intervals (CI) and plot the confidence interval from the estimates obtained from the B samples. The predictive ability of an estimate can be checked if it lies within the upper and lower limit of the CI. In the present study, similar bootstrap analysis is carried out (details are described in the following comment) and 95% CI for 50 and 100-year period quantiles, and the actual values are calculated at various thresholds and plotted as shown in Figure 9. The authors have also included the 95% CI of 50 and 100 year return period quantiles for LP 3 distribution in the revised manuscript. In this plot, the threshold obtained from the proposed entropy approach is compared to the performance of the statistical model fitted at the optimum threshold that would have been chosen based on goodness-of-fit metrics (Figure 7).

- I also do not understand how the bootstrapping was performed: could you provide a number for the length of data used for each resampling (line 344)?

  The non-parametric bootstrap sampling procedure is applied in the study;
  - $N_b$ bootstrapped series ($N_b = 1000$) of $X_p$ peaks are obtained by bootstrapping (i.e., resampling with replacement) of $X_p$ original peaks derived at each threshold level. So bootstrapped series are given by $\{X_p\}_j$ with j = 1, 2, 3…, $N_b$, and p is the number of flood exceedances at each threshold level; i.e., each bootstrapped series has an equal number of peaks as that of the original sample.
  - For each bootstrapped series $\{X_p\}_j$, distribution parameters, and 50, 100 -year flood quantiles are estimated.
  - The values of the estimates for a 95% confidence interval are calculated and plotted.

  These description has been included in the revised manuscript.

- In addition, the authors state at line 365 that the proposed method "gives more accurate optimum threshold values". Based on what facts do they claim the threshold identified from the entropy metric to be more accurate? What is their reference value?

  Considering the entropy of model 1, i.e., the arrival of peaks instead of taking only the entropy of distributions used for modeling exceedances gives more accurate optimum threshold values. The study involves the evaluation of an additional criterion to the goodness of fit of theoretical distribution calibrated to magnitude exceedances above the

threshold. It's observed that the threshold with the maximum entropy of Model 2 (i.e., used for the magnitude of exceedances) is different from the one at which the total entropy of both the models is the maximum (Figure 6). The latter one, selected as the optimum threshold, is close to other values obtained from some existing pieces of literature.

- Lines 358-363 simply discuss thresholds identified by means of alternative methods. If the value from the operational guidelines of Lang et al. (1999) (i.e., 730 m$^3$/s, line 359) is used as reference (although this is also just another method) then Langbein (1949) would still provide a more accurate threshold (716 m$^3$/s) than the proposed method (710 m$^3$/s). Please clarify.

  Here, the authors compare only the threshold values obtained from other existing guidelines or previous studies. They have compared only the values of optimum threshold obtained from different methods are close to the one proposed in the study. Like for LP 3 distribution, model 2 has the maximum entropy at a threshold of 900 m$^3$/s, while combing model 1 with this, the total entropy is maximized at a threshold of 710m$^3$/s, which is close to the values obtained from some existing methods. So considering the entropy of model 1 improves the accuracy of threshold identification for the proposed method. A similar explanation is added in detail in the revised manuscript.

  Additional points

- The proposed approach involves several steps which rely on visual observations and graphical analyses. These usually imply a high degree of subjectivity and difficulties to apply them to large datasets. It occurred to me that the approach described in section 2.4 to identify independent peaks is the same adopted by recent papers which leverage the Metastatistical Extreme Value framework to estimate flood magnitude and frequency from the whole series of ordinary peaks (i.e., with no need to define a threshold). Given that this novel statistical approach is gaining momentum, and that differently from the approach proposed here it can be completely automatized, it may be good to spend some words to justify the importance of identifying partial duration series by means of the classical peak over threshold methods.

  The proposed approach involves identifying an initial threshold range from the operational guidelines proposed by (Lang et al., 1999), which involves visual observation of mean residual life plot (MRLP). Domain 3 of Figure 3 can be identified once we extract independent peaks at each threshold level starting from the minimum daily discharge and calculate the length of the PDS. A visual inspection is needed for the MRLP to identify a possible range of thresholds where the optimum threshold might lie. Before the mere visual inspection, the numerical results obtained for plotting both the graphs should also be analyzed.

  The application of the metastatistical framework is adopted by some recent researchers to estimate flood magnitude and frequency from the whole series of ordinary peaks. However, the classical peak over threshold method is still popular among researchers while modeling with discharge data samples, and it involves a few uncertainties that need more attention. So in this work, the classical PDS sampling approach is further explored applying the concept of entropy. The application of this entropy-based concept can also be examined in the Metastatistical Extreme Value framework, which is out of the scope of this technical note. However, the author would like to address it in future research work along with the automation of the proposed entropy-based method.

Some suggestions concerning the structure of the paper:

- More precise explanations of what is shown in the figures and how it enables to reach the stated results are needed. Just to give two examples: line 240: how did Kendall's Tau verified the independence of the series? How do we see it? line 286: why finally the Poisson and not the Binomial distribution is chosen for the arrival of events above a threshold?

  The detailed procedure of Kendall's Tau test and Poisson hypothesis test is included in the revised manuscript. PDS at those thresholds where Kendall's tau is less than the critical value at 5% significance level are considered to satisfy the independence criteria. Similarly, based on the upper and lower limit of the dispersion index value, Poisson or Binomial distribution is selected. Details of these are described in the revised manuscript.

  Figures 1 to 5 display results of standard procedures which could be easily summarized with a few words in the text. Although this is a Technical Note where technical details shall be provided, Figure 2b-d only shows examples of results for arbitrarily chosen thresholds and Figure 4b is simply a zoom of Figure 4a. These figures could be deleted, which would help highlighting the actual results of the approach proposed in the paper (Figure 6).

  The authors have incorporated such modifications of figures in the main manuscript.

- Figures with several panels could be condensed. For example, Figure 6 could be condensed to Figure 6f only; Figure 7 can be condensed in one single panel displaying total rank only.

  The authors acknowledge this suggestion, and Figure 6 is modified in the revised manuscript. In Figure 7, the ranking of distributions at four different thresholds is shown for illustrative purposes. It's been changed in a single panel in the revised manuscript.

- Several minor issues exist in the paper, especially related to correct and precise use of language (e.g., 22, 68, 163, 167, 174, 128-129, 130, 212, 216, 255), definition of symbols and units (symbols shall be introduced the first time a variable is named, e.g., t is only defined at line 274 although appearing in Figure 1), differences between statements on the same subject (e.g., lines 88 and 314), motivations for showing these specific plots, given that many are examples for, e.g, different threshold (e.g., Figure 2 and 7). I do not detail them all here given the prior need to address the major issues described above. A carefully revision of the manuscript is however recommended.

  The authors would like to thank the reviewer for suggesting these corrections. Based on this, the manuscript is thoroughly revised to address all these minor issues related to language, symbols, units, etc.

**References**

Lang, M., Ouarda, T. B. M. J. and Bobe´e, B: Towards operational guidelines for over-threshold modeling, J, Hydrol, 225, 103–117, 1999.